# Scaling Multimodal Pre-Training via Cross-Modality Gradient Harmonization

**Junru Wu**[*]
Texas A&M University
sandboxmaster@tamu.edu

**Yi Liang**
Google Research
yiliang@google.com

**Feng Han**
Google Research
bladehan@google.com

**Hassan Akbari**
Google Research
hassanak@google.com

**Zhangyang Wang**
University of Texas at Austin
atlaswang@utexas.edu

**Cong Yu**[*]
Celonis Inc. / CeloAI
cong.yu@celonis.com

## Abstract

Self-supervised pre-training recently demonstrates success on large-scale multimodal data, and state-of-the-art contrastive learning methods often enforce the feature consistency from cross-modality inputs, such as video/audio or video/text pairs. Despite its convenience to formulate and leverage in practice, such cross-modality alignment (CMA) is only a weak and noisy supervision, since two modalities can be semantically misaligned even they are temporally aligned. For example, even in the (often adopted) instructional videos, a speaker can sometimes refer to something that is not visually present in the current frame; and the semantic misalignment would only be more unpredictable for the raw videos collected from unconstrained internet sources. We conjecture that might cause conflicts and biases among modalities, and may hence prohibit CMA from scaling up to training with larger and more heterogeneous data. This paper first verifies our conjecture by observing that, even in the latest VATT pre-training using only narrated videos, there exist strong gradient conflicts between different CMA losses within the same sample triplet (video, audio, text), indicating them as the noisy source of supervision. We then propose to harmonize such gradients during pre-training, via two techniques: (i) cross-modality gradient realignment: modifying different CMA loss gradients for one sample triplet, so that their gradient directions are in more agreement; and (ii) gradient-based curriculum learning: leveraging the gradient conflict information on an indicator of sample noisiness, to develop a curriculum learning strategy to prioritize training with less noisy sample triplets. Applying those gradient harmonization techniques to pre-training VATT on the HowTo100M dataset, we consistently improve its performance on different downstream tasks. Moreover, we are able to scale VATT pre-training to more complicated non-narrative Youtube8M dataset to further improve the state-of-the-arts.

## 1 Introduction

Self-supervised pre-training scales up deep learning to leverage massive unlabeled data. Beyond the maturity of pre-training over single-modality data such as language [2] or images [3], the recent success on multimodal pre-training [4, 5] demonstrates the versatility to synergize the rich multimodal information and to benefit a variety of downstream tasks. Many methods of this category [6–8, 4], especially the latest contrastive pre-training methods [5, 9–11], consider the most organic supervision as the *cross-modality alignment* (**CMA**), e.g., in the same video sequence, the video frames, audio

---

[*]Work done at Google Research

36th Conference on Neural Information Processing Systems (NeurIPS 2022).

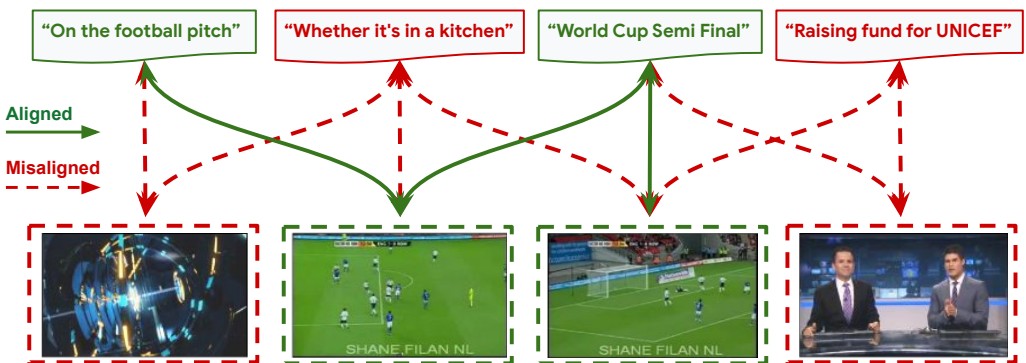

Figure 1: Examples of cross-modality alignment and misalignment existing in the complicated and non-narrative Youtube8M dataset [1]. Here we show a video sequence whose theme is a football match, while in some of its clips: (a) the text mentions content unrelated to the football, e.g., "kitchen","raising fund"; and (b) the visual content could also be unrelated to the football, e.g., TV cutscenes, the scene of the broadcast studio, etc.

stream, and text scripts are temporally aligned and hence should naturally have correspondence. Such assumed cross-modal correspondence could be exploited as "mutual supervision" for each modality to learn consistent semantic features with others. For example, the latest Video-Audio-Text Transformer (VATT) [11] hinges on this multimodal CMA *a priori* to train a transformer-based architecture (even a single backbone by sharing weights among three modalities), using two contrastive losses to align video-text and video-audio pairs, respectively. Their results set many new records in downstream tasks with neither supervised pre-training nor predefined inductive biases, showing strong "universal" generalizability despite the modality and domain gaps. In this paper, we focus on studying gradients in a modality-agnostic single-backbone setting of VATT [11].

Unfortunately, the CMA assumption rests on a very shaky foundation, especially for uncurated videos collected in the wild. The CMA can be weak even for the instructional videos such as HowTo100M [12] that are commonly adopted as the current multimodal pre-training data source. For instance, a speaker can refer to something before or after actually demonstrating it visually; she or he could also skip verbally explaining something that is visually happening, because that may be trivial or be already clear enough in the visual context [13]. Such discrepancy would be amplified if considering many other outliers in video (e.g. background objects) or language (e.g. off-topic small talks). The semantic misalignment will be even more severe, and in unpredictable forms, when we go beyond instructional videos and explore training with non-narrative, free-from internet videos. Figure 1 shows a few examples of such cross-modal misalignment in the non-narrative Youtube8M dataset [1], which would put CMA-guided pre-training in jeopardy. This important problem is yet under-studied in the self-supervised multimodal pre-training literature. That said, there has been a few recent efforts such as Multiple-Instance Learning (MIL) NCE [13] and noisy pair pruning as in [14].

We conjecture that the vanilla CMA might provide only weak, noisy and even misleading supervision for multimodal pre-training; hence it might constitute a hidden bottleneck when scaled up with more realistic data (beyond instructional videos) whose modalities are poorly aligned. We verify our conjecture by inspecting two large-scale multimodal datasets, HowTo100M [15] and Youtube8M [1]. Specifically, when we adopt pairwise CMA losses (e.g. video-audio, video-text) within the cross-modal sample triplets (video, audio, text), a majority of the gradients strongly conflict with each other in terms of their directions. We further identify (in Figure 3) that such strong conflict in gradients are correlated to noisiness of samples, indicating the noisy and misaligned supervision could be an important cause of gradient conflicts. Moreover, the cross-modal conflict leads to not only unstable training convergence, but also modality-specific overfitting; e.g. semantically strong representations for one modality while collapsed representations for another [16], yielding weak overall performance in cross-modality tasks such as Text-Video Retrieval as shown in Table 1.

Build upon the above conjectures and observations, we propose to harmonize those gradients so that they become mutually compatible in learning the unified representations. Specifically, we introduce two techniques: (i) cross-modal gradient realignment, where we modify different pairwise CMA loss gradients to align their directions for the same sample triplet, using gradient surgery [17] developed for multi-task learning; and (ii) gradient-based curriculum learning, where we leverage the gradient conflict information to indicate sample noisiness (e.g. a triplet whose CMA gradients are in more

agreement is considered less "noisy" and would be prioritized for cross-modal pre-training) and a curriculum learning strategy is developed accordingly. Both techniques are found to boost VATT performance on downstream tasks, not only on instructional videos, but even more when the more complicated non-narrative data is involved.

Our contributions can be summarized as the following:

- We suggest that the commonly assumed CMA might be an important yet overlooked performance hurdle for scaling up multimodal pre-training, and we observe severe misalignment even from the state-of-the-art (SOTA) model [11] on the relatively aligned data [12].

- We propose to consistently improve the pre-training pipeline in the only baseline in this direction, VATT (modality-agnostic setting), resulting in better downstream performance.

- With the help of our proposed techniques, we consistently improve the pre-training of VATT, resulting in better downstream performance, e.g. up to 58% gain in the rank metric of video-text retrieval task. Moreover, we are able to scale VATT pre-training to more complicated non-narrative Youtube8M dataset, which further yields new state-of-the-art performance in a modality-agnostic setting.

## 2 Related Work

### 2.1 Self-Supervised Multimodal Pre-training

The most organic self-supervision can arguably be found in the multimodal data that are abundantly available in the digital world: their temporal alignment naturally lead to cross-modality mutual supervision, hence meaningful features can be learned without requiring human annotation. Multimodal pre-training can thus leverage richer semantic cues than single-modality counterparts, and outperform in various downstream tasks such as image classification [5, 11], video action recognition [5, 11, 13, 10], audio event classification [5, 11, 10], and video-text retrieval [5, 11, 13]. Most of those methods formulate pre-text tasks, including cross-modality correspondence [6–8, 10], cross-modality clustering [18], cross-modality longer context prediction [19], or a combination of multiple pre-text tasks [20].

Contrastive learning [3, 21] has recently emerged as one dominant approach in self-supervised learning. In multimodal pre-training, the temporal co-occurrence naturally supplies the positive sample to contrast with. [5] adopts a multimodal versatile network to embed each modality into the same vector space, that is trained from end to end with multimodal pairwise contrastive losses (video-audio, and video text). CrossCLR [9] further takes the intra-modality similarities into account. [10] generalizes the instance discrimination idea to design a cross-modal discrimination task, i.e., predicting which audio matches a video. Most recently, VATT [11] combines the strength of convolution-free Transformer and multimodal contrastive learning. Specifically, it first validate the idea of using a modality-agnostic single-backbone transformer, to work on video, audio and text modalities, it follows the exact BERT [22] and ViT [23] architecture, except injecting modality-specific tokenization layers and linear projections.

Despite their success, existing multimodal contrastive learning methods [5, 9, 11] hinge on the "free" cross-modalities correspondence mined through the multimodal contrastive loss, i.e., the CMA assumption. As a result, they are often trained with well curated or narrated video datasets, such as AudioSet [24], YouCook2 [15], and HowTo100M [12]. It remains questionable whether such CMA-guided contrastive learning can generalize well to larger-scale multimodal data in the wild. In fact, even in the HowTo100M dataset, the authors of [12] estimate that around 50% of clip-narration pairs are not well aligned. The validity of CMA has arisen concerns. Most related to us is [13] which explicitly models the misalignment noise using the MIL-NCE loss, that is inherited by VATT [11].

### 2.2 Multi-task versus Multimodal Learning

While earlier self-supervised works used different modality-specific encoders, the latest multimodal pre-training sees a tendency to explore a versatile and "universal" model shared across modalities [5, 11]. It is easy to notice the resemblance between multimodal learning (especially with a unified backbone) and multi-task learning [25, 26]. The later often assumes multiple tasks to share transferable knowledge and can help each other learn. Yet in practice, different tasks can be heterogeneous in nature too, often leading to heavily misaligned gradient scales [27] or directions [17]. Such conflicting gradients from multiple losses would lead to several undesirable effects, including (i) strong bias

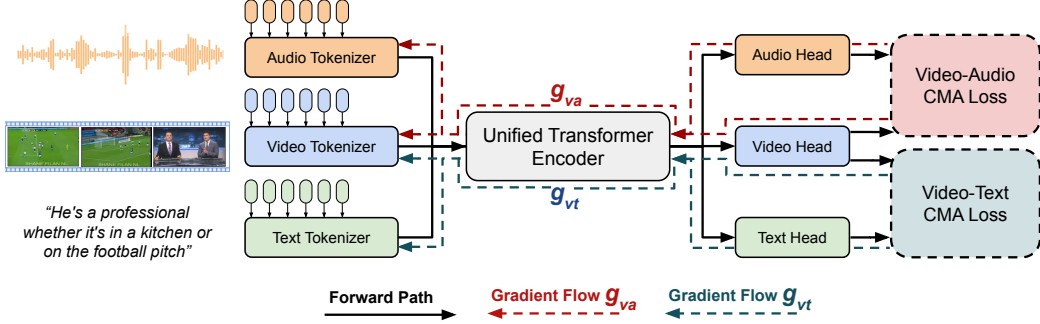

Figure 2: One main model used in this paper: the VATT model [11] with a modality-agnostic, single-backbone Transformer by sharing weights among the three modalities. VATT is trained with two contrastive losses between video-audio and video-text pairs, respectively. The positive cross-modality pairs to contrastive with are based on their temporal co-occurrence (e.g., the CMA assumption). We primarily study the conflicts between gradients $g_{va}$ and $g_{vt}$, which indicate the often poor alignments, and discuss how to harmonize their conflicts.

towards learning some tasks with largest gradient scales; and (2) slow and unstable convergence. Similar problems are often faced by multimodal pre-training too.

As a result, several works have been developed to mitigate the gradient dilemma in multi-task learning. Methods such as gradient normalization [27] and adaptive loss weight [28, 29] address the gradient scale/learning speed imbalance issues. Gradient surgery [17] tries to project gradients with conflicting directions onto the normal plane of each other, to prevent each gradient's interfering component from being applied to the network. Later, Gradient vaccine [30] further generalizes this idea extending the gradient projection to arbitrary angle, and uses the exponential moving average of cosine similarity to measure the alignment between different tasks. When it comes to the multimodal learning, [31] investigates optimally blending modality-specific gradients by linearly re-scaling them. Meanwhile, to our best knowledge, there has been no prior study on modality-specific gradient directions.

## 3    Methodology

**Pre-requisites:**  Without loss of generality, we focus on the SOTA multimodal contrastive learning model, VATT [11], as our default subject of study; and we follow its modality-agnostic single-backbone setting due to its compelling performance-efficiency trade-off, and due to the fact that it is the most intuitive setting to modify gradients (shared between paired modalities) w.r.t different end-point objectives.

In our modality-agnostic single-backbone setting of VATT, the goal of pre-training is to find the parameter $\theta$ of the model $f_\theta$ to learn meaningful features for all target modalities (video, audio, text), usually measured by a set of pairwise similarity metrics. VATT uses two pairwise contrastive losses for video-audio and video-text respectively to solve cross-modal alignment:

$$\min_\theta \ \text{CMA}_{va}(\theta) + \text{CMA}_{vt}(\theta) \tag{1}$$

where $\text{CMA}_{va}$ and $\text{CMA}_{vt}$ penalize the cross-modal alignment for video-audio and video-text pairs, using the Noise-Contrastive Estimation objective [32], respectively.

Figure 2 illustrates the VATT pipeline. First, video-audio-text triplets are sampled from random temporal locations, and then video-text and video-audio pairs are formed accordingly. Positive pairs are formed by selecting the two modalities at the same temporal locations from the same video clip; while the negative pairs are obtained by randomly sampling any video, audio, or text from other video clips. We follow the convention in [5, 11] to use the vanilla NCE loss for video-audio pairs, and to use the MIL-NCE loss proposed in [13] to align video-text pairs. Hence, $\text{CMA}_{va}$ and $\text{CMA}_{vt}$ are defined as follows:

$$\text{CMA}_{va}(\theta) = -\log\left(\frac{\exp(z_v^\top z_a/\tau)}{\exp(z_v^\top z_a/\tau) + \sum_{z'\in\mathcal{N}}\exp(z_v'^\top z'_a/\tau)}\right) \tag{2}$$

$$\text{CMA}_{vt}(\theta) = -\log\left(\frac{\sum_{z_t\in\mathcal{P}_k(z_t)}\exp(z_v^\top z_t/\tau)}{\sum_{z_t\in\mathcal{P}_k(z_t)}\exp(z_v^\top z_t/\tau) + \sum_{z'\in\mathcal{N}}\exp(z_v'^\top z'_t/\tau)}\right) \tag{3}$$

where $\mathcal{N}$ is the pool of negative pairs, $\mathcal{P}_k(z_t)$ denotes the $k$-neighbouring narrations surrounding the original text $z_t$. The gradients of two pairwise losses $\text{CMA}_{va}$ and $\text{CMA}_{vt}$, denoted as $g_{va}$ and $g_{vt}$, will be together applied (i.e. simple adding) to updating the model $f_\theta$.

### 3.1 Observation of Conflicting Gradients

While qualitatively recognizing the cross-modal misalignment is easy, it is a highly challenging endeavor to quantify the misalignment and to automatically find misaligned triplets at scale, due to the vaguely defined criterion as well as the absence of labels [13, 14]. Therefore, the foremost need is to identify a *surrogate indicator* on how noisy the alignment of a sample triplet is, without making unrealistic assumptions nor incurring much computational overhead.

Our empirical choice of surrogate indicator is the directional alignability (i.e., vector angle) of gradients generated by different pairwise cross-modal losses, i.e., the **cosine similarity** between $g_{va}$ and $g_{vt}$ in the particular example of VATT, after flattening them into two vectors. The choice is firstly rationalized by the *coherent gradient hypothesis* [33], that healthy model training should witness the overall gradient stronger in certain directions along which the samples' gradients reinforce each other. Secondly, comparing the gradient directions between multiple pairwise losses could be considered as an ensemble or "cross-check": intuitively, if both video-audio and video-text pairwise consistency lead to the same update direction, then there is a good chance that those modalities are well aligned and the update direction is reliable; otherwise, at least one pair (video-audio, or video-text) might suffer from misalignment and provides noisy cross-modality guidance.

Overall, we conjecture that: *aligned video-text-audio triplets should have higher cosine similarity for $g_{va}$ and $g_{vt}$, and vice versa.*

To further validate the conjecture, we conduct a sanity check, where we started from a pre-trained VATT network and further optimize it for 200 iterations with a batch size of 64, on Youtube8M dataset. We then randomly sample video-text-audio triplets out of the samples with top 5% and bottom 5% most aligned gradient, measured by $cos(g'_{va}, g'_{va})$. In Figure 3, we visualize 10 frames within a 32-frame video clip and its corresponding text narration (including 8 neighbouring text narrations similar to [13]). We visually observed that the top 5% group triplets has more semantically aligned words in the corresponding text narration (highlighted in **green**) thus enjoy a better cross-modality alignment, while the bottom 5% group triplets has fewer semantically aligned words, therefore are much more noisy in their alignments. We include more visualization samples in the Appendix.

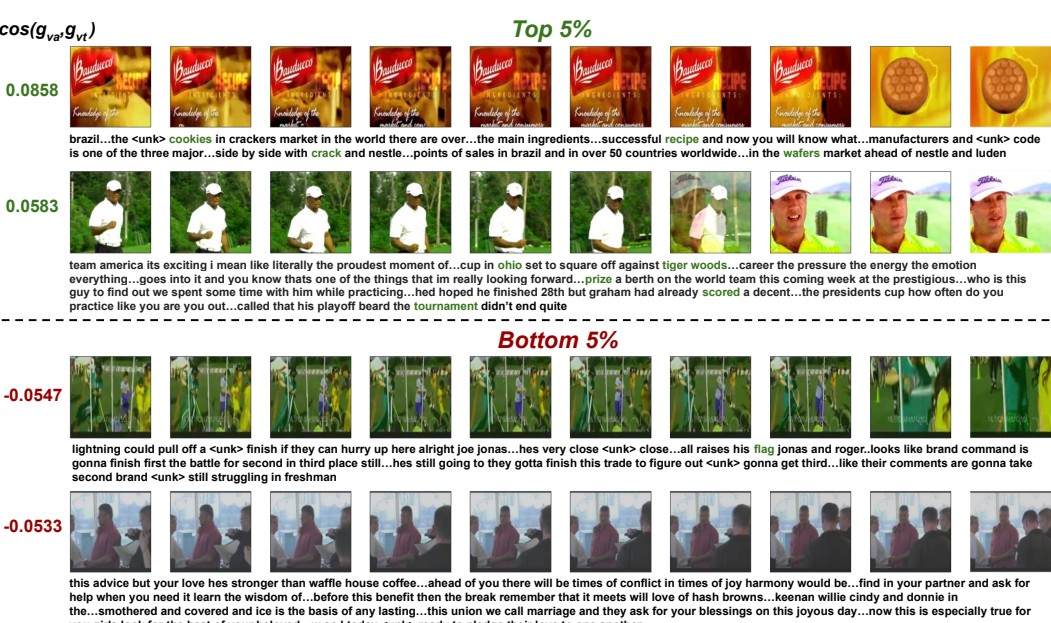

Figure 3: Qualitative example of our proposed measure based on agreement of gradients on Youtube8M dataset, semantically aligned words are highlighted in **green**, $cos(g_{va}, g_{vt})$ reflect the agreement of between $g_{va}$ and $g_{vt}$, by measuring their cosine similarity.

With the assumption that misalignment in gradients correlates to the noisiness of the sample's cross-modality alignment, we use the former as the computable surrogate indicator of the later, and examine the cosine similarity between gradient $g_{va}$ and $g_{vt}$ (over 500k iterations) of VATT on the HowTo100M dataset [12]. In Figure 4a, we plot the distribution of cosine similarities, $cos(g_{va}, g_{vt})$, across 500k iterations of pre-training. We observe that $cos(g_{va}, g_{vt})$ at any iteration resembles a normal distribution, and about half of the gradients $g_{va}$ and $g_{va}$ have misaligned directions (negative cosine similarities). The plot echos the empirical finding in [15] that around 50% of video-text pairs are not well-aligned on the HowTo100M dataset. In the Appendix, we have included similar plots for the YouTube8M dataset [1]. Comparing those plots verify that non-narrative datasets have much worse misalignment than narrative ones, hence challenging the CMA assumption even more.

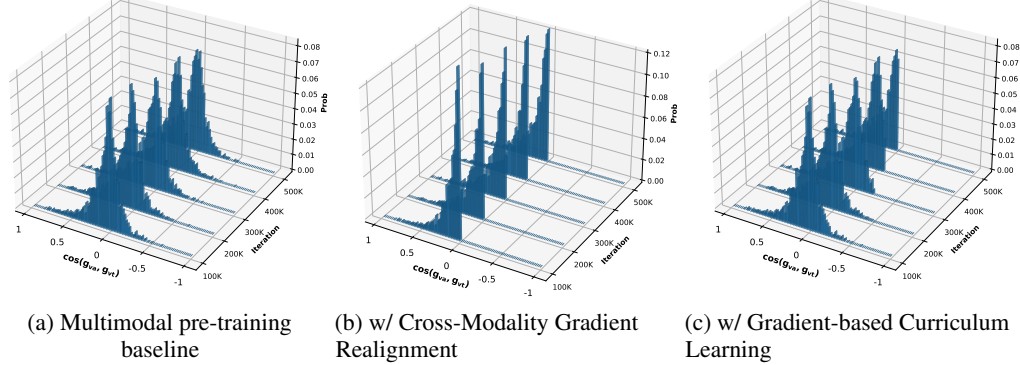

(a) Multimodal pre-training baseline

(b) w/ Cross-Modality Gradient Realignment

(c) w/ Gradient-based Curriculum Learning

Figure 4: Visualization of cosine similarity between gradient $g_{va}$ and $g_{vt}$ across 500K Iterations on the HowTo100M dataset. (a) shows the baseline setting in multimodal pre-training; (b) with only cross-modality gradient realignment; (c) with only gradient-based curriculum learning. Youtube8M follows very similar observations, see Appendix.

The existence of conflicting gradients not only makes the convergence harder, but also makes the learned representation heavily overfit some modalities. In our case, we observe the training to be biased towards video/audio modalities, which favors the downstream performance of video/audio classification task, yet sets back the performance of video-to-text retrieval task (verified in Table 1).

## 3.2 Harmonizing Gradients during Pre-training

We present two strategies to mitigate the conflicting gradients: a "hard" way to project gradient directions, and a "soft" way to progressively select sample triplets in curriculum.

### 3.2.1 Cross-Modality Gradient Realignment

---
**Algorithm 1** Cross-Modality Gradient Realignment
---
**Require:** Model parameter $\theta$ , minibatchs $\mathcal{B}_{va}$, minibatchs $\mathcal{B}_{vt}$
  **for** $(b_{va}, b_{vt}) \in (\mathcal{B}_{va}, \mathcal{B}_{vt})$ **do**
    $g_{va} \leftarrow \nabla_\theta \text{CMA}_{va}(\theta), g_{vt} \leftarrow \nabla_\theta \text{CMA}_{vt}(\theta)$
    $g_{va} \leftarrow \text{flatten}(g_{va}), g_{vt} \leftarrow \text{flatten}(g_{vt})$
    $\hat{g}_{va} \leftarrow g_{va}, \hat{g}_{vt} \leftarrow g_{vt}$
    **if** $g_{va} \cdot g_{vt} < 0$ **then**
       $\hat{g}_{va} \leftarrow \hat{g}_{va} - \frac{\hat{g}_{va} \cdot g_{vt}}{||g_{vt}||^2}$      ▷ projection $g_{vt} \leftarrow g_{va}$
       $\hat{g}_{vt} \leftarrow \hat{g}_{vt} - \frac{\hat{g}_{vt} \cdot g_{va}}{||g_{va}||^2}$      ▷ projection $g_{va} \leftarrow g_{vt}$
    **end if**
    $\hat{g}_{va} \leftarrow \text{reshape}(\hat{g}_{va}), \hat{g}_{vt} \leftarrow \text{reshape}(\hat{g}_{vt})$
    $\Delta\theta \leftarrow \hat{g}_{vt} + \hat{g}_{va}$      ▷ sum up gradients
    update $\theta$ with $\Delta\theta$      ▷ update parameter
  **end for**
---

Since $g_{va}$ and $g_{vt}$ are misaligned in terms of their directions, one natural idea is to re-align the cross-modality gradients, by enforcing or re-projecting their directions to be aligned. An intuitive choice is an effective remedy developed for multi-task learning named Gradient Surgery [17]. Here, instead of applying it to $g_i$ and $g_j$ where $i, j$ are the two heterogeneous tasks, we apply it to $g_{va}$ and $g_{vt}$, which refer to the gradients of the two pairwise losses $\text{CMA}_{va}$ and $\text{CMA}_{vt}$ w.r.t the weights in the model, following our rationale in Section 3.1.

If the gradients $g_{va}$ and $g_{vt}$ have negative cosine similarity, we alter them by projecting each onto the normal plane of the other. Otherwise, if $g_{va}$ and $g_{vt}$ have zero or positive similarity, we retain the original gradients. We refer to this particularly adapted form of gradient surgery as *Cross-Modality Gradient*

*Realignment.* Its update procedure is outlined in Algorithm 1, and the overhead very cheap, mainly just computing inner products of two flattened gradients. We also tried another similar algorithm named Gradient Vaccine [30], and the results are almost identical to using Gradient Surgery [17]. We hence only report one for the sake of conciseness.

### 3.2.2 Gradient-based Curriculum Learning

We next propose a gradient-based curriculum learning strategy that gradually identifies and removes more misaligned samples based on the similarity indicator as the training goes on. Formally, given a video-audio pair $b_{va}$ and video-text pair $b_{vt}$, we only update the parameter if $cos(g_{va}, g_{vt}) > \gamma$ and drop this sample triplet otherwise. This ensures us to gradually prioritize training with well-aligned samples while suppressing the noisy supervision from potentially misaligned samples. The drop rate here is controlled by a cut-off threshold $\gamma$. In general, $\gamma$ is a negative value monotonically increasing during training, and there is an underlying rationale.

Initially, the network is under-trained and the similarity between its learned features is not reliable to indicate a strong CMA; hence we are more conservative in excluding samples. As the network becomes more sufficiently trained, the features become more semantically aligned, allowing us to remove more misaligned samples in the later stage of training. The full procedure of Gradient-based Curriculum Learning is outlined in Algorithm 2. We empirically choice $\gamma_0 = -0.3$ at the beginning and linearly increase it to $\gamma_{500K} = 0$ at the end, we include

---

**Algorithm 2** Gradient-based Curriculum Learning

---

**Require:** Model parameter $\theta$, minibatchs $\mathcal{B}_{va}$, minibatchs $\mathcal{B}_{vt}$, initial $\gamma_0$
  **for** $(b_{va}, b_{vt}) \in (\mathcal{B}_{va}, \mathcal{B}_{vt})$ **do**
    update $\gamma$           ▷ curriculumly update $\gamma$
    $g_{va} \leftarrow \nabla_\theta \text{CMA}_{va}(\theta)$, $g_{vt} \leftarrow \nabla_\theta \text{CMA}_{vt}(\theta)$
    $g_{va} \leftarrow \text{flatten}(g_{va})$, $g_{vt} \leftarrow \text{flatten}(g_{vt})$
    **if** $g_{va} \cdot g_{vt} > \gamma$ **then**
      $g_{va} \leftarrow \text{reshape}(g_{va})$, $g_{vt} \leftarrow \text{reshape}(g_{vt})$
      $\Delta\theta \leftarrow g_{va} + g_{vt}$      ▷ sum up gradients
      update $\theta$ with $\Delta\theta$      ▷ update parameter
    **end if**
  **end for**

---

an ablation on the adaptive choice of $\gamma$ in the Table 2.

Similar classes of ideas, i.e, "gradually focusing a small training subset", have been explored by a prominent class of algorithms which rely on sample selection to robustify [34–36] or to accelerate [37] training. Most of them use the "small loss trick" wherein a fraction of samples with smallest loss values below a certain threshold are considered as reliable. However, they did not explore gradient conflict indicators, which targets solving our CMA problem here. Also note that our two steps can fully reuse the cosine similarity computed, so applying them together incurs no extra overhead.

## 4 Experiments

### 4.1 Pre-Training Datasets and Settings

In this paper, we focus on studying gradients in a modality-agnostic single-backbone setting, thus we do not consider multimodal pre-training baseline with modality-specific models, e.g. HERO[38], MMV[5], PerceiverIO[39], AVLnet[40]. Apart from VATT[11], there exist other concurrent work that use modality-agnostic setting[20, 41, 42], yet they all have their respective limitations. For example, UniVL[20] lacks audio modalities thus cannot constitute video-audio CMA loss, VLM[41] applies separate attention mask for each modality while PolyViT[42] use alternate training between modalities and tasks, thus eliminate the possibilities of any gradients conflicts. Therefore, this leave VATT[11] to be the most suitable baseline besides it SoTA performance.

Our pre-training adopts three settings for comprehensive comparison: **(1)** using HowTo100M as our training set, following [20, 12]; **(2)** combining HowTo100M and AudioSet as our training set following [5] and VATT [11]; and **(3)** additionally merging a noisy Youtube8M dataset alongside with HowTo100M and AudioSet, to create a much more compound and complicated training set than considered before, to stretch-test our proposal's benefits to scale up pre-training in uncurated and poorly aligned data.

For each setting, we i.i.d sample video-audio-text clips from the mixture of all candidate sets in every batch. (HowTo100M, Audioset or Youtube8M, if applicable). For all setting, we only use a subset of HowTo100M, AudioSet, Youtube8M, Kinetics400 and Youcook2 dataset in compliance with Youtube's wipe-out policies.

**HowTo100M**[1][12] is a large-scale dataset of narrated videos with an emphasis on instructional videos that have well synchronized modalities. The training set consists of 127 Millions video clips with the corresponding close-captioned text from automatic speech recognition (ASR).

**AudioSet**[1][24] is a large-scale audio-visual dataset originally intended for audio event classification. It consists of 1.78 Million 10-second clips sampled from Youtube Videos that contains a variety of audio tracks, such as musical instruments, animals or mechanical sounds.

**Youtube8M**[1][1] is a large-scale video classification dataset based on Youtube videos. It consists of a diverse vocabulary of 3862 visual entities and a total number of 8 Million videos. To the best of our knowledge, we are the first to scale contrastive multimodal pre-training beyond instructional videos, onto this noisy Youtube8M videos that more faithfully represent videos the wild. For data pre-processing, we split each video into 5s clips and use ASR close-captions to generate the corresponding text, resulting in 165 Millions video-audio-text clips.

## 4.2 Pre-Training Settings

**Network Architecture:** For all of our experiments, we use modality-agnostic Transformer variant in [11], VATT-MA-Medium. Specifically, we use a 12-layer Transformer, in which each layer has a feed-forward MLP with hidden size of 4096, and 16 attention heads with a hidden size of 1024.

**Pre-training Hyperparameter:** We strictly follow the setting in [11], pre-training VATT from scratch with Adam optimizer with an initial learning rate of 1e-4, 10k warmup steps, 500k steps in total, a batch size of 2048 and using a cosine learning rate scheduler to anneal the learning rate from 1e-4 to 5e-5. Our framework is implemented in Tensorflow 2.8, and train with 256 TPUV3s, it took a total of 3 days to train our models.

## 4.3 Downstream Tasks for Evaluation

We evaluate the performance of our pre-trained representations when tuned towards various downstream tasks, including a variety of video/audio and cross modalities tasks.

**Video Action Recognition:** Following [5], we evaluate on UCF101[43] (101 action classes, 13,320 videos) and the HMDB51[44] (51 classes, 6,766 videos) benchmarks. For both settings, we directly attach a linear classifier on the learned representations while freezing the transformer backbone. We also evaluate on Kinetics400[1][45] dataset (400 classes, 234,584 video), where we fine-tune the transformer backbone.

**Audio Event Classification:** Following [5], we used ESC50[46] (50 classes, 2000 audio clips) as the benchmark for audio modalities. Similarly, we directly attach and tune a linear classifier on top of the frozen transformer backbone.

**Zero-shot Video-Text Retrieval:** To evaluate our performance on cross-modality task, we follow [5] using YouCook2[1][15] (3.1K video-text clips) and MSR-VTT[1][47] (1K video-text clips) as our evaluation datasets. We directly use the embedding from the pre-trained model without any fine-tuning, and report the recall at 10 (R@10) and Median Rank as described in [5].

## 4.4 Result Analysis on HowTo100M and AudioSet Pre-training

**Gradient Re-weighting** We set up a simple gradient re-weighting baseline similar to [31], where we simply scale up the gradient magnitude of $g_{va}$ by 2.5 times and scale down the gradient magnitude of its counterpart $g_{vt}$ by 0.5 times, we denote this baseline as Gradient Re-weighting (VA), we also did it vice versa and denoted it as Gradient Re-weighting (VT). As shown in Table 1, we did not observe much performance boost with gradient re-weighting, on the contrary, there are significant performance drop in Text-Video Retrieval task on both YouCook2 and MSRVTT datasets.

**Cross-Modality Gradient Realignment** Compared to Gradient Re-weighting, our Cross-Modality Gradient Realignment mitigates the gradient conflicts by modifying gradients direction. In Table 1, we can see significant performance gains in video-text retrieval task (e.g. 58% gain in rank metric on YouCook2, 46% gain in R@10 metric on MSR-VTT), while maintaining a comparable performance in video/audio tasks. Since video-text retrieval task heavily hinge on cross-modality alignment (CMA), this improvement verify the benefits of re-aligning the cross-modality gradients.

---

[1]It is important to note that Howto100M, Youtube8M, AudioSet, Kinetics400, MSRVTT and YouCook2 videos are sourced from YouTube. Many of the videos have been made explicitly unavailable [48], hence we only train and evaluate over the subset of data that is publicly available at the time.

| Dataset | Tasks | Video Action Cls | | | | | | Text-Video Retrieval | | | | Audio Cls | |
|---|---|---|---|---|---|---|---|---|---|---|---|---|---|
| | Dataset | UCF101 | | HMDB51 | | Kinetics400 | | YouCook2 | | MSRVTT | | ESC50 | |
| | Metric | Top1↑ | Top5↑ | Top1↑ | Top5↑ | Top1↑ | Top5↑ | Rank↓ | R@10↑ | Rank↓ | R@10↑ | Top1↑ | Top5↑ |
| HT100M | VATT [11] | 78.53 | 95.24 | 57.36 | 86.07 | 74.71 | 92.69 | 93.50 | 17.10 | 73.00 | 16.70 | 71.50 | 91.75 |
| | + RW (VA) | 78.24 | 95.01 | 58.74 | 85.39 | 70.55 | 90.41 | 168.90 | 9.24 | 100.20 | 13.56 | 71.56 | 93.02 |
| | + RW (VT) | 78.03 | 95.35 | 58.23 | 86.80 | 75.66 | 92.80 | 90.35 | 19.26 | 62.50 | 18.02 | 71.29 | 91.38 |
| | + GR | 78.44 | 95.37 | 54.38 | 83.64 | 76.72 | 92.72 | 47.00 | 24.94 | 68.00 | 19.20 | 69.00 | 93.00 |
| | + CL | 79.10 | 96.16 | 56.41 | 86.06 | 77.25 | 93.38 | 40.00 | 26.01 | 56.00 | 23.90 | 72.50 | 94.25 |
| | + Both | 79.24 | 96.58 | 58.24 | 87.37 | 76.59 | 93.26 | 42.00 | 25.87 | 54.00 | 24.50 | 72.05 | 94.16 |
| HT100M + AudioSet | VATT [11] | 84.40 | - | 63.10 | - | 79.23 | 94.30 | 34.00 | 29.00 | 67.00 | 23.60 | 81.20 | - |
| | + RW (VA) | 83.44 | 97.28 | 59.55 | 88.02 | 76.56 | 93.52 | 238.50 | 6.80 | 147.00 | 12.30 | 81.75 | 97.25 |
| | + RW (VT) | 84.42 | 97.49 | 62.30 | 86.61 | 78.59 | 94.17 | 76.00 | 18.72 | 120.00 | 15.20 | 80.75 | 96.75 |
| | + GR | 84.77 | 97.38 | 62.30 | 90.38 | 79.29 | 94.32 | 29.00 | 31.65 | 70.00 | 21.40 | 81.50 | 97.00 |
| | + CL | 86.04 | 97.75 | 65.45 | 88.94 | 79.89 | 94.71 | 33.00 | 29.17 | 65.50 | 20.97 | 82.00 | 97.25 |
| | + Both | 85.46 | 97.58 | 65.52 | 89.74 | 79.26 | 94.48 | 31.50 | 30.26 | 69.50 | 19.96 | 82.00 | 97.00 |
| HT100M + AudioSet + YT8M | VATT [11] | 88.28 | 98.73 | 65.84 | 91.43 | 79.39 | 94.56 | 29.00 | 29.66 | 56.00 | 26.90 | 80.75 | 97.00 |
| | + RW (VA) | 86.97 | 98.09 | 61.06 | 89.66 | 77.70 | 93.83 | 99.00 | 14.25 | 75.50 | 19.70 | 83.50 | 97.25 |
| | + RW (VT) | 88.19 | 97.96 | 61.13 | 90.51 | 78.43 | 94.38 | 27.00 | 31.07 | 48.50 | 27.70 | 82.25 | 96.75 |
| | + GR | 87.49 | 98.10 | 60.99 | 88.35 | 79.73 | 94.57 | 32.00 | 29.56 | 60.00 | 27.20 | 85.00 | 98.00 |
| | + CL | 89.02 | 98.33 | 65.77 | 92.15 | 79.70 | 94.80 | 31.00 | 31.34 | 48.50 | 28.70 | 83.50 | 97.75 |
| | + Both | 89.70 | 98.35 | 64.35 | 92.08 | 80.01 | 94.69 | 29.00 | 31.86 | 45.00 | 29.10 | 84.50 | 98.00 |

Table 1: Results of pre-training on HowTo100M, AudioSet and Youtube8M. *Best* results are highlighted in **blue**, *second best* results are highlighted in **light blue**. RW(VA): Gradient Re-weighting (Scale up Video-Audio), RW(VT): Gradient Re-weighting (Scale up Video-Text), GR: Cross-Modality Gradient Realignment, CL: Gradient-based Curriculum Learning. (best view in color)

| Hyperparameter | | Video Action Cls | | | | Text-Video Retrieval | | | | Audio Cls | |
|---|---|---|---|---|---|---|---|---|---|---|---|
| Dataset | | UCF101 | | HMDB51 | | YouCook2 | | MSRVTT | | ESC50 | |
| Metric | | Top1↑ | Top5↑ | Top1↑ | Top5↑ | Rank↓ | R@10↑ | Rank↓ | R@10↑ | Top1↑ | Top5↑ |
| VATT [11] | | 78.53 | 95.24 | 57.36 | 86.07 | 93.50 | 17.10 | 73.00 | 16.70 | 71.50 | 91.75 |
| + CL | $(\gamma_0, \gamma_{500K}) = (-0.3, 0.0)$ | 79.10 | 96.16 | 56.41 | 86.06 | 40.00 | 26.01 | 56.00 | 23.90 | 72.50 | 94.25 |
| | $(\gamma_0, \gamma_{500K}) = (-0.2, 0.0)$ | 77.75 | 94.94 | 56.87 | 87.18 | 53.00 | 22.21 | 57.00 | 20.80 | 70.75 | 94.75 |
| | $(\gamma_0, \gamma_{500K}) = (-0.1, 0.0)$ | 44.49 | 75.42 | 33.44 | 68.78 | 81.00 | 17.13 | 68.50 | 19.10 | 70.25 | 93.25 |
| | $(\gamma_0, \gamma_{500K}) = (0.0, 0.0)$ | 10.05 | 24.88 | 7.31 | 23.76 | 457.50 | 5.02 | 153.50 | 11.20 | 58.50 | 87.50 |

Table 2: Ablation on the choice of $\gamma$ in Gradient-based Curriculum Learning on Howto100M. We linearly decay $\gamma$ during training, $\gamma_0, \gamma_{500K}$ denote $\gamma$ at the beginning and at the end of the training, respectively. *Best* results are highlighted in **blue**. CL: Gradient-based Curriculum Learning.

**Gradient-based Curriculum Learning** Compared to Cross-Modality Gradient Realignment, Gradient-based Curriculum Learning inherents its performance gains in video-text retrieval task, yet enjoy an additional performance boost in audio classification (e.g. 1.40% gain ESC50 dataset in Table 1) and video classification (e.g. 1.94% gains UCF101 dataset in Table 1). We also perform an ablation on the adaptive choice of $\gamma$ on Howto100M dataset in Table 2. We can see that curriculumly evolving $\gamma$ (i.e. $(\gamma_0, \gamma_{500K}) = (-0.3, 0.0)$) yields the best overall results, while maintaining a fixed $\gamma$ (i.e. $(\gamma_0, \gamma_{500K}) = (0.0, 0.0)$) yields the worst results, this verify the effectiveness of curriculumly removing mis-aligned samples during training.

**Cross-Modality Gradient Realignment + Gradient-based Curriculum Learning** We further verify the effectiveness with the combination of both techniques. Specifically, given a video-audio pair and video-text pair, (a) if $cos(g_{va}, g_{vt}) \leq \gamma$, we drop this sample triplet; (b) if $\gamma < cos(g_{va}, g_{vt}) < 0$, we apply gradient projection in Algorithm 1; (c) if $cos(g_{va}, g_{vt}) \geq 0$, we retain the original gradients. We found they are complimentary to each other and the combination of both lead to better performance in the majority of downstream tasks.

### 4.5 Scaling Pre-training to YouTube8M

As shown in Table 1, adding YouTube8M dataset brings the video classification and video-text retrieval performance to a new level (e.g. 88.28% Top1 on UCF101, 56.0 rank on MSRVTT)), yet it set back the performance in audio classification task, we conjecture this is largely due to the noisy misaligned samples introduced by Youtube8M dataset.

However, by leveraging our proposed techniques, we further yields a new state-of-the-art performance in the modality-agnostic setting, without any modification in architecture, striking 89.70% Top1 on

UCF101, 92.15% Top5 on HMDB51, 80.01% Top1 on Kinetics400 and 85.00% Top1 on ESC50 dataset, demonstrating the effectiveness of our method when scaling to noisy Youtub8M dataset.

## 4.6 Feature Visualization

To better illustrate *why gradient harmonization techniques works*, in Figure 5, we follow the visualization used in VATT[11], which take YouCook2 Video-Text Retrieval dataset, showing the output space visualization of video and text, respectively. We can see the initial video and text feature representation of VATT baseline are mixed together (which aligned with the findings in [11]), however, after using gradient realignment, curriculum learning or applying both, the video and text representations become disentangled, this indicates our gradient harmonization techniques adds to cross-modality diversity, which makes video/text more discriminative from each other, making the model more discriminative and better in generalization.

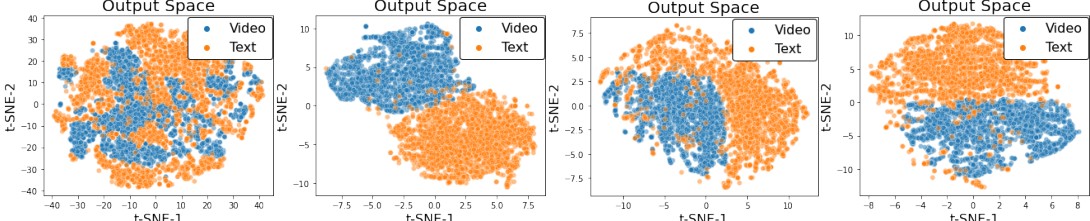

(a) VATT Baseline     (b) w/ Gradient Realignment(c) w/ Curriculum Learning     (d) w/ Both Techniques

Figure 5: t-SNE visualization of the VATT output space on YouCook2 dataset.

## 4.7 Extending Gradient Realignment to Modality-Specific VATT

So far, our techniques were built upon the modality-agnostic setting of VATT with a shared backbone. We also try to apply the same technique to the modality-specific setting, given there is still a shared video head where we can harmonization the gradients. In Table 3, we observe that applying Gradient Realignment (GR) to modality-specific setting has some gain over uni-modality tasks (e.g. Video/Audio Classfication), yet set back on cross-modality tasks (e.g. Text-Video Retrieval). We hypothesize this is due the lack of cross-modality signals: since there is no longer shared backbone across modalities, the only shared video head would provide bias signala towards the video modality, for cross-modality alignment. We leave the improvement of modality-specific VATT for future work.

| Methods | Video Action Cls | | | | Text-Video Retrieval | | | | Audio Cls | |
|---|---|---|---|---|---|---|---|---|---|---|
| Dataset | UCF101 | | HMDB51 | | YouCook2 | | MSRVTT | | ESC50 | |
| Metric | Top1↑ | Top5↑ | Top1↑ | Top5↑ | Rank↓ | R@10↑ | Rank↓ | R@10↑ | Top1↑ | Top5↑ |
| Modality-Specific VATT [11] | 82.36 | 97.35 | 58.51 | 89.32 | 19.00 | 38.84 | 27.00 | 32.38 | 74.72 | 93.75 |
| + GR | 82.06 | 97.35 | 61.45 | 90.45 | 22.00 | 36.82 | 40.00 | 27.32 | 76.50 | 93.00 |

Table 3: Extended experiments on applying Gradient Realignment to the modality-specific VATT.

## 5 Conclusion

In this paper, we take a deep dive into the common CMA assumption used in multimodal pre-training. Using the gradient directions between pairwise losses as the surrogate indicator, we observe ubiquitous gradient conflicts even in relatively well-aligned narrative datasets. We then propose two plug-and-play techniques to harmonize the gradients during training. They are demonstrated to substantially enhance VATT pre-training across a number of downstream tasks, and further scale it up to pre-training with even more complicated and poorly aligned data. Our findings underline the (often overlooked) importance to carefully revisit the basic assumption and training dynamics of multimodal learning, besides advancing the model architectures. However, our cross-modality gradient harmonization technique still has room to generalize better to the modality-specific setting. Additionally, we did not fully explore other regularization techniques in multi-tasking learning literature, such as [27–29]. The potential negative societal impacts include the malicious use of multimodal models, such as unintended use of its downstream visual/audio recognition capability.

## Acknowledgments and Disclosure of Funding

We give special thanks to Tianqi Liu and Jialu Liu for the constructive feedback and discussions. Z. Wang is supported in part by US Army Research Office Young Investigator Award W911NF2010240.

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
