# Scaling Multimodal Pre-Training via Cross-Modality Gradient Harmonization Suppmentary Material

## 1 Qualitative examples of our proposed measure based on agreement of gradients

To further validate the conjecture that *aligned video-text-audio triplets should have higher cosine similarity for $g_{va}$ and $g_{vt}$, and vice versa*, we conduct a sanity check, where we started from a pre-trained VATT network and further optimize it for 200 iterations with a batch size of 64 on Youtube8M dataset, we then randomly sample video-text-audio triplets out of the samples with top 5% and bottom 5% most aligned gradient, measured by $cos(g'_{va}, g'_{va})$. In Figure 1 and 2, we visually observed that the top 5% group triplets has more semantically aligned words in the corresponding text narration (highlighted in **green**) thus enjoy a better cross-modality alignment, while the bottom 5% group triplets has fewer semantically aligned words, therefore are much more noisy in their alignments.

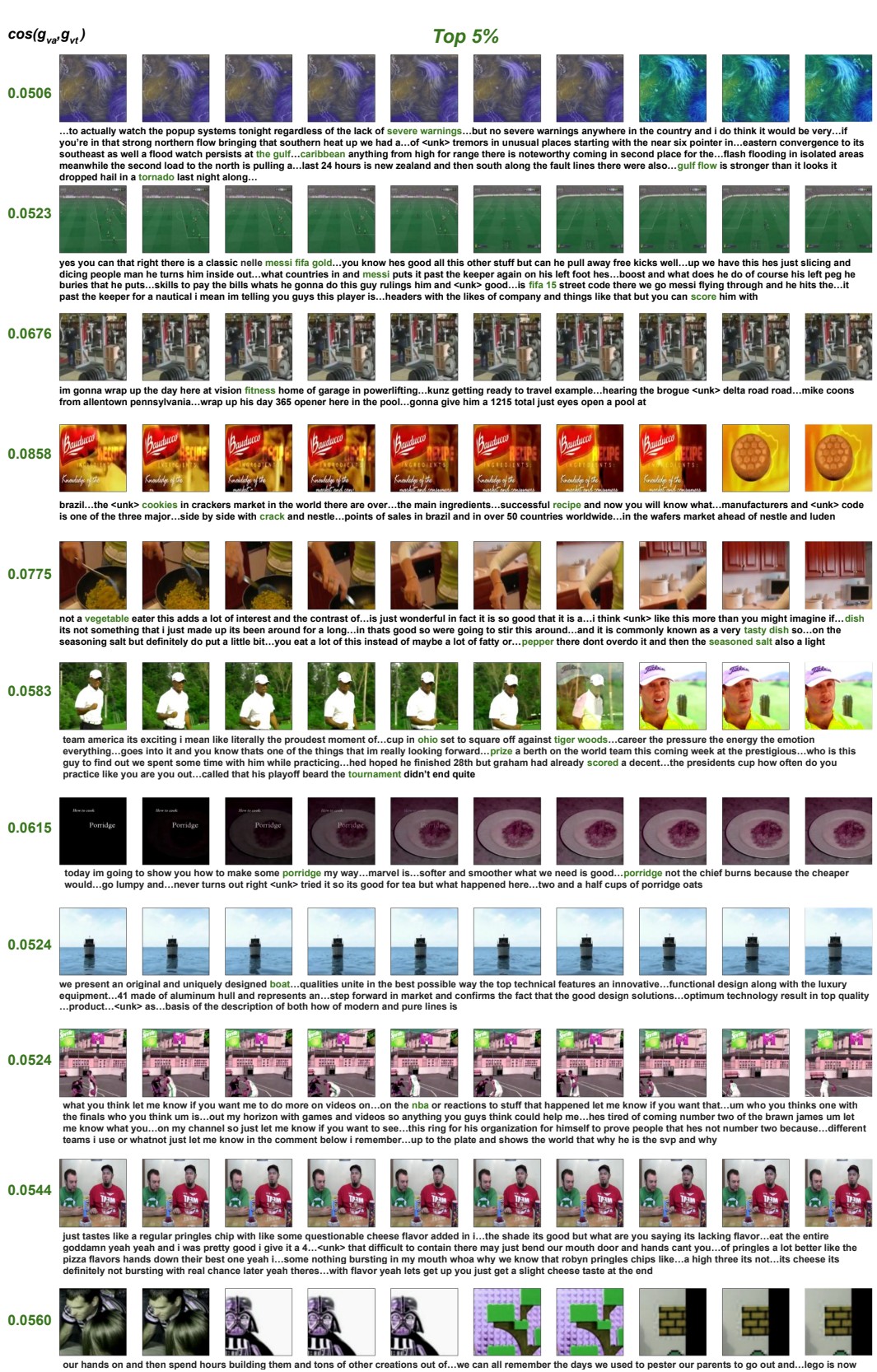

Figure 1: Visualization of Top 5% examples measured by the agreement of gradients on Youtube8M dataset, semantically aligned text are highlighted in **green**, $cos(g_{va}, g_{vt})$ reflect the agreement of between $g_{va}$ and $g_{vt}$, by measuring their cosine similarity.

$cos(g_{va}, g_{vt})$

**-0.0473** however existing road signs were totally inadequate for the new…was booming in the government alarmed at the clogged up roads decided to build the first…highways no one had designed motorway signs before because we…<unk> work began in the 1950s when car…had a motorway before and the whole job of making that system of…important than i really am i doubt it…clear to the car which was still a new thing remains me and at speeds that…all fell to a man and his former student from an art college they gave you…perhaps maybe

**-0.0458** kind of doing your gs this way also…so you can so you can…yeah its good over and…so yeah shes just for it…into that jesus…and then you can also pay this…you just play this its like a million f major seven

**-0.0507** to a 20 in the series it started out fairly stalled a little bit or pace…nexus they will and in under 24 minutes shall goo were gonna go up a…out of the way cacao is up in 10 seconds shall <unk> may have enough to…have at the first exit are will fall the second does not look particularly healthy as…as the copper behind them ig may have just traded two members of xiao do for…shower get themselves an age and they have a massive mini way to cannock creeps as…and that is cleaning houses at i the only member of ig left alive will fall…of damage swift who died back in hell get the kill…but ill come to their…hair down there for kitty so im you get a ride in up to cacao will

**-0.0459** seconds still down continuing into this tight <unk> still down…the short rise because its not worth putting it up then over the bridge and…except for a <unk> climb in there somewhere its settled down for the next…here for to ops and a rough downhill sweeper and tight turns…it up for only the third time in three minutes here just before this steep and some more climbing as you can see a dropper post can give you gains just…for the climb then <unk> got some flat trail then the saddle down again…everywhere it doesnt have to be downhill here are my top three…look for wide lines on such a narrow trail the saddle goes

**-0.0547** lightning could pull off a <unk> finish if they can hurry up here right alright joe jonas…hes very close <unk> close…all raises his flag jonas and roger..looks like brand command is gonna finish first the battle for second in third place still…hes still going to they gotta finish this trade to figure out <unk> gonna get third…like their comments are gonna take second brand <unk> still struggling in freshman

**-0.0487** up on the outside he <unk> moved but hes coming into it strongly and further back…then mr jackman open book followed by barbed as held in a pocket <unk> is…zoo stars about the claim them cassidy had a look then i look to the left…and <unk> is racing away with it at the hundred meter mark zoo start…bella ed lost in was referred little miss smiley led from var penn zero…putting a gap in them here hands and heels only wins it by four on barbed…shes got them covered the dominant philly beat sensibility bound for earth in arabian gulf…mr jackman followed by penn zero then havana flak jacket and little miss…grid one sensibility sticking on on the outside but quilts to good despite sitting wide

**-0.0533** this advice but your love hes stronger than waffle house coffee…ahead of you there will be times of conflict in times of joy harmony would be…find in your partner and ask for help when you need it learn the wisdom of…before this benefit then the break remember that it meets will love of hash browns…keenan willie cindy and donnie in the…smothered and covered and ice is the basis of any lasting…this union we call marriage and they ask for your blessings on this joyous day…now this is especially true for you girls look for the best of your beloved…w co l today <unk> ready to pledge their love to one another

**-0.0477** it goes on to baby care which and then you delicates then <unk> got your daily…know looks at what you your load is that <unk> put into machine…which you can choose a really quick wash on that if you want to and i…is friendly on how much water it uses and…wash that it does on here is a <unk> wash when we get…need on this machine and you go here from the super eco wash…to this lcd display you can choose your temperatures so…watches and i cant think of anything else that you…when you select from any one of these programs which is

**-0.0502** i would definitely wear something like this one of my own wedding functions…it is me and i think he he…so fine and feminine and elegant and i …sees maybe look good today and im babe i wake up i always believe that theres…everything that he does is his structure is very sharp but the the workmanship is…of people doing by the way im very heavy by the way but no ones really…about her grandmas clothes is that he maintains a traditional aesthetic…way for people who are going to thailand to get married or going to who <unk>…stunning in that beautiful attire i think experimentation is great but i what what i truly

**-0.0502** cut these molds real easily ill finish packing this mold and ill let these…and be really careful if you scrape away the clay like this you…pieces dry ill dry these either in the refrigerator or just air dry now to…with the excess im going to pack that into the antique mold to make that little…the two pieces i mix up some thick copper clay paste you can follow our…pick and i hold that straight up and down so i get a nice straight edges …on making bronze clay pastes its the exact same method…cut out the cushion shape that i need using my ultra…then i need to position it right in the center

**-0.0477** there and theres the top right there as exceeds black and great yeah…definitely love the gorilla logo so on japanese with the little top…got some nice little detailing on what not on the top and bottom oh you can…copper color and yeah so overall it looks like a…what kind of can its supposed to be representing but it looks cool are definite like…of nice representation of a can of some sort of…back to the can i can see see here the kind of a soda can the…thing very cool now to get this into…really keen yeah now

Figure 2: Visualization of Bottom 5% examples measured by the agreement of gradients on Youtube8M dataset, semantically aligned text are highlighted in **green**, $cos(g_{va}, g_{vt})$ reflect the agreement of between $g_{va}$ and $g_{vt}$, by measuring their cosine similarity.

## 2 Training dynamics of our proposed measure on Youtube8M dataset

In Figure 3, we plot the distribution of cosine similarities, $cos(g_{va}, g_{va})$, across 500k iterations of pre-training, on the noisy Youtube8M dataset. We observe that $cos(g_{va}, g_{va})$ at any iteration resembles a normal distribution, and about half of the gradients $g_{va}$ and $g_{va}$ have misaligned directions (negative cosine similarities). We also calculate the mean of $cos(g_{va}, g_{va})$ across all 500k iterations on Youtube8M dataset, and found it to be 30% smaller than that on Howto100M dataset, indicating stronger mis-alignment in gradient directions on Youtube8M, which further verify that non-narrative video sets such as Youtube8M have more severe misalignment than the narrative ones such as Howto100M, hence challenging the CMA assumption commonly used in multimodal pre-training[**? ?** ].

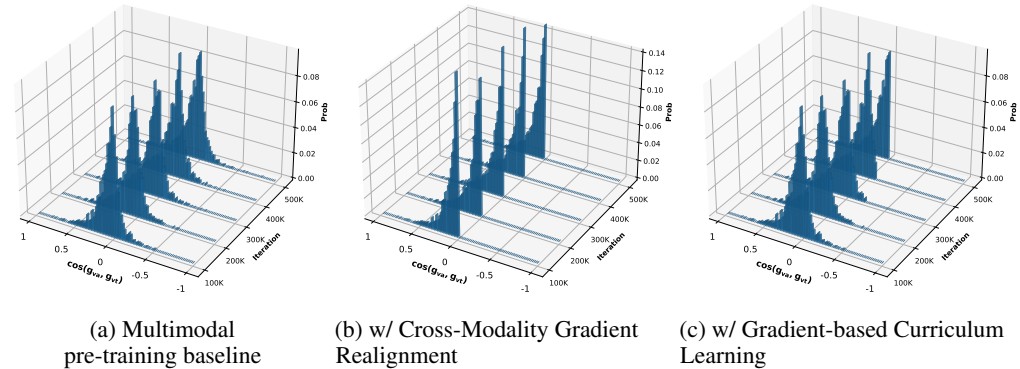

(a) Multimodal
pre-training baseline

(b) w/ Cross-Modality Gradient
Realignment

(c) w/ Gradient-based Curriculum
Learning

Figure 3: Visualization of cosine similarity between gradient $g_{va}$ and $g_{vt}$ across 500K Iterations on the Youtube8M dataset. (a) shows the baseline setting in multimodal pre-training; (b) with only cross-modality gradient realignment; (c) with only gradient-based curriculum learning.