# OpenReview forum: "Scaling Multimodal Pre-Training via Cross-Modality Gradient Harmonization"
_NeurIPS.cc/2022/Conference — NeurIPS 2022 Accept_

### Official Review · Reviewer_LhYL · 2022-07-10

**Rating:** 7
**Confidence:** 4
**Soundness:** 3 good
**Presentation:** 4 excellent
**Contribution:** 4 excellent

**Summary:**

This paper challenges the cross-modality alignment (CMA) assumption in multi-modal pre-training. The authors found that CMA can be weak and noisy because of the often poor cross-modality semantic alignment. They proposed two techniques, i.e., gradient realignment and gradient-based curriculum learning. Experiments on conducted with VATT, on some of the large-scale pre-training datasets.



**Questions:**

Please see the weakness part

**Limitations:**

No, the current version claimed it has no negative social impacts.

**Strengths And Weaknesses:**

Strengths:
-	The authors studied a novel and important problem. CMA is traditionally assumed by default in multi-modal learning (and kinda why most prior works was restricted to instructional video data). This paper takes an important step toward the wild.

-	The observation of poor CMA is convincing (Figure 1 illustration, and Figure 3 quantification by gradients). This problem of misalignment has not been studied in the existing works on self-supervised multimodal learning, and the insight to analyze the problem from the perspective of the gradient is new to my best knowledge.

-	The two proposed techniques, albeit simple, seem effective and low overhead to implement

-	The authors spent lots of effort on experiments. The benchmarking in Tables 1 and 2 is comprehensive.

-	The authors are able to scale up VATT pre-training to the more complicated non-narrative Youtube8M dataset, which further yields new state-of-the-art performance in a modality-agnostic setting. This is impressive and validates the benefits of their proposed techniques.

Weaknesses:
-	The method has moderate technical novelty since it adapts several known wisdoms from multi-task learning.

-	This paper in general lacks an ablation study. I can totally understand the (re-)training overhead is huge for such models, but at least some rationale or comparison on choosing their methods’ ad-hoc hyperparameters (there are many) could be provided?

-	How might the proposed technique scale beyond VATT? E.g., to last V-L models such as DALL-E and CLIP? Do they also suffer from similar problems, and can your methods be helpful? (I’m not asking authors to re-train CLIP or alike, just idea sketching).

-	The writing quality of this paper is barely readable and needs proofreading.

---

> ### Author Response · Authors · 2022-08-02
> **Respond to Reviewer LhYL**
>
> **Q1: Comparison on choosing their methods’ ad-hoc hyperparameters.**
> - Thanks for pointing out, we are running more experiments on more ablation of learning rate and batch size, we will update the results in the rolling rebuttal period as soon as it’s ready.
>
> **Q2: Scaling beyond VATT, such as DALL-E and CLIP.**
> - DALL-E: DALL-E is a text-to-image autoregressive generative model to model the text and image tokens as a single stream of data, since the single stream of data consist of both text and image. Our method could potentially apply to this setting, given there exist gradient conflicts between $g_{txt}$ and $g_{img}$ within a single stream of data.
> - CLIP: CLIP use use separate image and text encoder during contrastive pre-training, thus our method cannot be directly applied to the original CLIP, however it could be applied to a modality-shared CLIP variant MS-CLIP [1], since it introduced a modality shared encoder, creating potential gradient conflicts between $(g_{img}, g_{txt})$. Therefore, we can then use our gradient harmonization, by directly applying our techniques to $(g_{img}, g_{txt})$.
>
> [1] You et al., MS-CLIP: Towards Modality-Shared Contrastive Language-Image Pre-training. ECCV 2022.
>
> **Q3: The writing quality of this paper can be improved**
> - Thanks for the suggestion, we will polish the writing in the final version.

---

### Official Review · Reviewer_CREH · 2022-07-10

**Rating:** 5
**Confidence:** 5
**Soundness:** 3 good
**Presentation:** 4 excellent
**Contribution:** 2 fair

**Summary:**

This paper proposes two regularization strategies to harmonize the multimodal pre-training. The motivation is that cross-modality alignment (CMA) is problematic when the pre-training data is noisy and misaligned. The contribution of this work is: (1) introducing the cosine similarity between video-audio and video-text gradients as a surrogate indicator on how noisy the alignment of a sample triplet (video-audio-text) is; (2) introducing cross-modality gradient realignment to re-project the gradient direction if they have negative cosine similarity; and (3) using cosine similarity to prioritize training with well-aligned samples. The effectiveness of the proposed strategies are comprehensively evaluated.


**Questions:**

1. Line 51: “Figure ??”. This should be a typo.
2. Table 1: when the pre-training datasets are HT100M and AudioSet, the performance of VATT on Text-Video retrieval is quite low. For example, this paper reports R@10=29 of VATT on YouCook2. However, the original VATT paper reports R@10=40.6 (the higher the better) for YouCook2.


**Limitations:**

Not applicable.


**Strengths And Weaknesses:**

---

Originality: misalignment is a well-known problem in multimodal pre-training data, but is also tricky to solve. The two regularization strategies proposed in this work are novel in the multimodal pre-training community.

---

Quality:

Strengths: using cosine similarity of video-audio and video-text gradients as the indicator of misalignment is reasonable. The effectiveness of the proposed regularizations are evaluated on various downstream tasks. The work is complete.

Weakness: (1) cosine similarity can only be applied to the scenario with at least triplet. When the pre-training data only contains video-text pairs or only contains video-audio pairs, this surrogate indicator does not work. (2) According to Table 1, the majority of the improvement in video action classification tasks comes from curriculum learning (CL). Combining with gradient realignment (GR) even hurts the performance of CL-only.

---

Clarity: this paper is well written and well organized. It is easy to follow. Since this work is built upon VATT and provides enough training details, it is easy for readers to reproduce the result.

---

Significance: misalignment is an important problem in contrastive-based representation learning. Although it is novel to use cosine similarity of two gradients to measure the misalignment, the performance improvement is marginal.

---

---

> ### Author Response · Authors · 2022-08-02
> **Respond to Reviewer CREH**
>
> **Q1: Cosine similarity can only be applied to the scenario with at least triplet?**
> - Our methods could actually be straightforwardly applied to settings where it only contains image-text pairs, for example, in the modality-shared CLIP variant MS-CLIP [1], their setting only have image-text pairs and a modality-shared encoder. Even so, we can still apply our gradient harmonization techniques to $g_{img}$ and $g_{txt}$ on the modality-shared encoder, despite that the physical meanings might be slightly different. As discussed in Sec. 3.1, for $cos(g_{va}, g_{vt})$, comparing the gradient directions between video-audio and video-text pairwise losses could be considered as an ensemble or “cross-check": intuitively, if both video-audio and video-text pairwise consistency agree on the update direction, then there is a good chance that those modalities are well aligned and the update direction is reliable; otherwise, at least one pair (video-audio, or video-text) might suffer from misalignment and CMA provides noisy guidance. In contrast, comparing $g_{img}$ and $g_{txt}$ alone won’t provide as reliable cross-check information, but can still be done.
>
> **Q2: According to Table 1, the majority of the improvement in video action classification tasks comes from curriculum learning (CL). Combining with gradient realignment (GR) even hurts the performance of CL-only.**
> - We hypothesize that gradient realignment and gradient-based curriculum learning might have different dynamics when updating gradients, i.e. curriculum learning does not distort gradient, it tend to learn samples curriculumly and put hard samples at last, hence is often the preferred option. On the other hand, gradient realignment need to distort gradients, which could be seen as a more aggressive way to align modalities. As a result, in most cases curriculum learning is more helpful than gradient realignment.
>
> - However, with the complicated non-narrative Youtube8M dataset, the gradients are more noisy (line 188-189), it is preferred to discard noisy gradients and realign distorted gradients simultaneously, to boost the cross-modality alignment. This also matches our empirical finding that when adding YouTube8M dataset, gradient realignment + curriculum learning are the most helpful in downstream cross-modality tasks (video-text retrieval), but less helpful (even hurt) on single modality tasks (video action classification). As a result, we hypothesize that such a combination works better with stronger gradient noise on downstream cross-modality tasks.
>
> **Q3: Typos**
> - Thanks for pointing this out! We have fixed those typos in the updated version.
>
> **Q4: The performance of VATT reported in the paper is lower compared to the original VATT paper?**
> - This is because the pre-training dataset (Howto100M, AudioSet) and evaluation dataset (YouCook2) are sourced from YouTube, many of the videos have been made explicitly unavailable [2], hence the size of dataset is always shrinking over time. For fair comparison, we re-trained and evaluated the VATT model on the up-to-date datasets, thus the re-trained VATT baseline (R@10=29) might sometimes not match the original VATT baseline (R@10=40.6) reported in the paper. The validity of our reproduction has been personally confirmed with VATT authors.
>
> [1] You et al., MS-CLIP: Towards Modality-Shared Contrastive Language-Image Pre-training. ECCV 2022.
>
> [2] Lucas Smaira, João Carreira, Eric Noland, Ellen Clancy, Amy Wu, and Andrew Zisserman. A short note on the kinetics-700-2020 human action dataset. arXiv preprint arXiv:2010.10864, 2020.

---

### Official Review · Reviewer_VTo5 · 2022-07-12

**Rating:** 7
**Confidence:** 4
**Soundness:** 3 good
**Presentation:** 2 fair
**Contribution:** 4 excellent

**Summary:**

This paper studies the problem of contrastive multimodal pre-training (assuming triplet training data <video, text, audio>). It starts with uncovering the misalignment phenomenon existing among temporally-associated video clips, subtitles, and audio tracks, which make them falsely considered positive training pairs. It then quantitatively measures the gradient conflicts resulted from this misalignment. Ideally, given a modality-agnostic backbone, gradient updates come from different loss terms on the same positive triplet should reinforce each other, reflected by non-negative cosine similarity. However, this is not the case for existing methods. To this end, the paper proposes two techniques, namely Gradient Realignment (forcing gradients updates from different losses to have non-negative cosine similarity) and Gradient-based Curriculum Learning (filter out samples with low gradient similarity), to alleviate the gradient conflicting issue.

**Questions:**

Typos:

i) Line 51, the figure reference is unknown.

ii) The arrow in Colume 6 Row 3 of Tab. 1 is missing.

**Limitations:**

Appear to be sufficient.

**Strengths And Weaknesses:**

The proposed idea is sensible and interesting. Overall, the experimental evaluation is comprehensive and supports the effectiveness of the proposed method.

The motivation of the idea, however, could be more upfront. As the paper suggests in the rather later part of the paper (line 255), the proposed method is only studied in the modality-agnostic single-backbone setting. However, without this knowledge, understanding the motivation of the paper (gradient conflicts) is cumbersome. Please consider stating the problem setting clearly as early in the paper as possible (on or before line 61).

Moreover, is the proposed method only applicable for modality-agnostic architecture with triplet training samples? How about the modality-specific setting, where the gradients $g_{va}$ and $g_{vt}$ conflict on the video encoder only, but not audio encoder and text encoder (as it is the pivot)? Or in the bi-modal modality-agnostic setting (e.g., image and text [r1]), should the gradients backprop from the image feature and text feature be harmonized? It would be great if the authors could share insights/analyses on whether the proposed method could tackle these scenarios.

The final question is related to the previous one. As $g_{va}$ consists of gradients from both the audio part and the video part, $g_{vt}$ consists of gradients from both the video part and the text part, should this "micro" harmonization between all four of them be considered in addition to the "macro" harmonization showcased in the paper? Any insights are appreciated.

[r1] You et al., MS-CLIP: Towards Modality-Shared Contrastive Language-Image Pre-training. ECCV 2022.

---

> ### Author Response · Authors · 2022-08-02
> **Respond to Reviewer VTo5**
>
> **Q1: The motivation of the idea, however, could be more upfront. Please consider stating the modality-agnostic setting clearly as early in the paper as possible (on or before line 61)**
>
> -  In the updated version of the paper, we add a mention of our modality-agnostic setting in the introduction (line 42-43).
>
> **Q2: Can our method generalize beyond triplet training pairs? E.g. modality-specific setting or bi-modal modality-agnostic setting**
> - Great question. Our method can also generalize to modality-specific settings, where we only need to harmonize the gradient conflicts between $g_{va}$ and $g_{vt}$ in the Video Head (Figure 2). We are running experiments on modality-specific settings, and we will update the results in the rolling rebuttal period as soon as it’s ready.
> - Our method also works for bi-modal modality-agnostic settings such as MS-CLIP[r1], Since in their settings, the gradients from image ($g_{img}$) and text ($g_{txt}$) go through a modality-shared encoder, creating potential gradient conflicts. We will cite MS-CLIP [r1] in the related work of modality-agnostic settings, we will also apply our method to MS-CLIP [r1] as soon as they have released the training code.
>
> [r1] You et al., MS-CLIP: Towards Modality-Shared Contrastive Language-Image Pre-training. ECCV 2022.
>
> **Q3: "macro" harmonization of $(g_{va}, g_{vt})$ v.s. "micro" harmonization of $(g_{v} , g_{a})$ in $g_{va}$ and $(g_{v}, g_{t})$ in $g_{vt}$ ?**
>
> - Great question. We can decouple $g_{va}$ into $g_{v}$  and $g_{a}$  by using stop gradient techniques (similarly $g_{vt}$ into $g_{v}$ and $g_{t}$), we can then further apply gradient harmonization techniques to gradient pairs $(g_{v} , g_{a})$ and $(g_{v}, g_{t})$ in additional to the existing gradient pair $(g_{va}, g_{vt})$.
> - However, $cos(g_{v} , g_{a})$ or $cos(g_{v} , g_{t})$ represents a different physical meaning compared to  $cos(g_{va} , g_{vt})$. As discussed in Sec. 3.1, for $cos(g_{va} , g_{vt})$, comparing the gradient directions between video-audio and video-text pairwise losses could be considered as an ensemble or “cross-check": intuitively, if both video-audio and video-text pairwise consistency agree on the update direction, then there is a good chance that those modalities are well aligned and the update direction is reliable; otherwise, at least one pair (video-audio, or video-text) might suffer from misalignment and CMA provides noisy guidance. In contrast, comparing $g_{v}$ and $g_{a}$ (or $g_{t}$) alone won’t provide as reliable cross-check information, but can still be done.
>
> **Q4: Typos i) Line 51, the figure reference is unknown. ii) The arrow in Colume 6 Row 3 of Tab. 1 is missing**
> - Thanks for pointing this out! We have fixed those typos in the updated version.

---

> > ### Comment · Reviewer_VTo5 · 2022-08-08
> > **Thanks for the responses**
> >
> > Thanks for the clarification and insights. Look forward to seeing these additional results in the final version of the paper. (Note that MS-CLIP has just released their code: https://github.com/Hxyou/MSCLIP)

---

> > > ### Author Response · Authors · 2022-08-09
> > > **Respond to Reviewer VTo5**
> > >
> > > Thanks for the acknowledgement, we will add additional results on MS-CLIP [r1] in the final version of the paper, as soon as they have the pre-training code released.

---

### Meta-Review · Area_Chair_WKGs · 2022-08-21

**Recommendation:** Accept
**Confidence:** Certain

**Metareview:**

Authors present a mechanism to improve triple cross-modality alignment between video-text / video-audio with a shared encoding backbone. The basic idea is to measure the cosine similarity between the gradients coming from video-text and video-audio. When the gradients conflict, the authors postulate that this is caused by mis-matched data. In order to address, the gradients can either be re-aligned, or this information can be used for curriculum learning to filter noisy samples.

Authors study on 6 datasets, modifying VATT with their approach, and demonstrate consistent improvements in performance.


Pros:
- [R] Well written (disagreement) and easy to follow.
- [R] Interesting method with clear impact.
- [R] Experimental evaluation is comprehensive
- [R] Method can scale up to noisier data (YouTube8M) to demonstrate even further gains in performance.

Cons:
- [AC] Authors assume that when gradients conflict, this correlates with noisy data. But no study was performed to confirm this. The method may improve performance on end tasks, but the motivation as to why it improves performance is only based on assumptions, and no data. Authors could greatly improve the paper by doing a study on a random sample of conflicting gradients to confirm that the data samples in those situations are misaligned more than when the gradients are aligned.
- [R] Only applies to modality agnostic single backbone setting. Authors should make this more clear in the paper. Authors have addressed this concern.
- [R] Writing could be improved.

Unanimous reviewer ratings on accept. In light of the utility of the approach, the measured improvements in performance, and reviewer comments, AC recommends accept. AC, however, recommends to authors to reframe their writing to emphasize that they *assume* gradient misalignment correlates to noisy samples, or perform an experiment to confirm that this is the case.

AC Rating: Accept

**Award:**

No

---

### Decision · Program_Chairs · 2022-09-14

Accept